# Genome-Wide Identification and Comprehensive Analysis of the GASA Gene Family in Peanuts (*Arachis hypogaea* L.) under Abiotic Stress

**DOI:** 10.3390/ijms242317117

**Published:** 2023-12-04

**Authors:** Rizwana Begum Syed Nabi, Myoung Hee Lee, Kwang-Soo Cho, Rupesh Tayade, Sungup Kim, Jung-In Kim, Min-Young Kim, Eunsoo Lee, Jungeun Lee, Sang-Woo Kim, Eunyoung Oh

**Affiliations:** 1Department of Southern Area Crop Science, National Institute of Crop Science, RDA, Miryang 50424, Republic of Korea; ruhii.syed@gmail.com (R.B.S.N.); kji1204@korea.kr (J.-I.K.);; 2Department of Applied Biosciences, Kyungpook National University, Daegu 41566, Republic of Korea; rupesh.tayade@gmail.com

**Keywords:** GASA, peanut, gibberellin, drought, salt stress, phylogenetic analysis, expression analysis

## Abstract

Peanut (*Arachis hypogaea* L.) is a globally cultivated crop of significant economic and nutritional importance. The role of gibberellic-acid-stimulated Arabidopsis (GASA) family genes is well established in plant growth, development, and biotic and abiotic stress responses. However, there is a gap in understanding the function of GASA proteins in cultivated peanuts, particularly in response to abiotic stresses such as drought and salinity. Thus, we conducted comprehensive in silico analyses to identify and verify the existence of 40 GASA genes (termed *AhGASA*) in cultivated peanuts. Subsequently, we conducted biological experiments and performed expression analyses of selected *AhGASA* genes to elucidate their potential regulatory roles in response to drought and salinity. Phylogenetic analysis revealed that *AhGASA* genes could be categorized into four distinct subfamilies. Under normal growth conditions, selected *AhGASA* genes exhibited varying expressions in young peanut seedling leaves, stems, and roots tissues. Notably, our findings indicate that certain *AhGASA* genes were downregulated under drought stress but upregulated under salt stress. These results suggest that specific *AhGASA* genes are involved in the regulation of salt or drought stress. Further functional characterization of the upregulated genes under both drought and salt stress will be essential to confirm their regulatory roles in this context. Overall, our findings provide compelling evidence of the involvement of *AhGASA* genes in the mechanisms of stress tolerance in cultivated peanuts. This study enhances our understanding of the functions of *AhGASA* genes in response to abiotic stress and lays the groundwork for future investigations into the molecular characterization of *AhGASA* genes.

## 1. Introduction

The gibberellic acid (GA)-stimulated *Arabidopsis* (GASA) gene family comprises a group of small cysteine-rich peptides that have been found in a wide range of plant species [1]. GA is a diterpenoid that is highly conserved in plants, and it plays a vital role in many physiological processes, such as cell division, plant development, and stress response. Since the identification of the first GASA gene in 1992, numerous genome-wide studies have been conducted on GASA genes in various plant species, including monocotyledonous and dicotyledonous plants, resulting in the isolation and identification of several GASA genes [2,3]. For instance, 15 GASA genes have been identified in *Arabidopsis* (*Arabidopsis thaliana*), whereas 18 have been identified in tobacco (*Nicotiana tabacum*). In maize, Zimmermann et al. (2010) [4] reported the existence of 10 GASA genes. Likewise, in rice (*Oryza sativa*), 10 GASA genes were identified [5]. Similarly, 37 GASA genes were identified in soybean (*Glycine max*), [6] whereas 21 and 19 genes were found in the poplar species *Populus trichocarpa* and *P. euphratica*, respectively [7,8]. GASA genes have multifaceted roles in growth and development and a wide variety of roles in regulating processes such as cell growth and division, biotic and abiotic stress responses, the transmission of signals from hormones, and redox reactions. Thus, in recent years, research on the GASA gene family has become increasingly common because of their potential applications in crop improvement [7,8,9,10,11,12]. For example, *OsGSR1* in rice is involved in the regulation of GA and brassinosteroid (BR) crosstalk through interactions with BR synthase DIM/DWF1 proteins [13]. Another study revealed the involvement of *TaGASR1* in heat stress regulation in wheat [14]. Similarly, *CrGASA*s were found to be involved in heat tolerance and heavy metal tolerance [15].

Peanut (*Arachis hypogaea* L.) is an important legume with significant agricultural and economic value globally, and it is widely recognized for its high nutrient, protein, and oil content and extensive use in various culinary and industrial applications [16,17]. Peanuts have become a staple food and cash crop in many regions. Peanuts are believed to have originated in South America, particularly in present-day Bolivia and Peru [17]. Over time, this crop spread to other parts of the world through exploration and trade routes. Currently, peanuts are grown in numerous countries across different continents, including the United States, China, India, Nigeria, and South Korea. The importance of peanuts as a cash crop lies in their versatility and adaptability. Peanuts can be grown in diverse climates, including arid and semi-arid regions in which other crops can struggle. The ability of the peanut plant to fix nitrogen in the soil through symbiotic relationships with nitrogen-fixing bacteria enhances soil fertility and contributes to sustainable agricultural practices.

Climate change is an unavoidable and major factor limiting crop yield. Similarly, drought and salinity are major abiotic stress factors that significantly impact the production of crops, including peanuts [18,19]. These stress factors can reduce plant growth and productivity by affecting various physiological and biochemical processes in plants [20,21]. In recent years, the identification and characterization of genes that play a crucial role in abiotic stress tolerance have become popular research topics in plant biology. In this context, researchers recently identified the genome-wide GASA gene family in the cultivated peanut, as well as its wild relatives, *A. duranensis* and *A. ipaensis*. Researchers have analyzed the distribution of these genes on chromosomes, their structural characteristics, and the differences between cultivated and wild peanuts. Researchers additionally assessed the expression of GASA genes during pod development [8]. As previously mentioned, the GASA gene family has been identified as an important regulator of plant growth and development under various environmental stress conditions. Thus, this study performed biological experiments on peanuts and evaluated the roles of selected GASA genes in the response to drought and salt stress. We further investigated the expression patterns of the identified peanut GASA genes under drought and salt stress. Additionally, we assessed the GASA gene sequences of peanut and other plant species and investigated their phylogenetic relationships. Our findings could shed light on the molecular mechanisms underlying the responses of peanut plants to abiotic stress and provide new insights for the development of stress-tolerant peanut varieties.

## 2. Results

### 2.1. Phylogenetic Analysis of the GASA Gene Family

To discover the evolutionary inter-relatedness of the AhGASA proteins, a phylogenetic tree was constructed using multiple sequence alignments of GASA proteins from peanut, *Arabidopsis*, tomato, soybean, and cotton. According to the phylogenetic tree (Figure 1), GASA genes can be categorized into four subfamilies, namely, subfamilies CI, CII, CIII, and CIV (Figure 1). *AhGASA* genes from peanut are distributed throughout the four subfamilies together with *Arabidopsis*, tomato, soybean, poplar, and cotton GASA genes. Subfamily CI (Figure 1) contains 43 GASA genes, including 9 *AhGASA* genes, 12 *GlymaGASA* genes, 11 *GohirGASA* genes, 4 *PotriGASA* genes, 5 *LOCOsGASA* genes, and 2 *ATGASA* genes. Subfamily CII contains the highest number of GASA genes (56), including 15 *AhGASA* genes, 4 *LOCOsGASA* genes, 6 *ATGASA* genes, 18 *GohirGASA* genes, 8 *GlymaGASA* genes, and 5 *PotriGASA* genes. Likewise, subfamily CIII contains 25 GASA genes, including 9 *GlymaGASA* genes, 7 *GohirGASA* genes, 6 *AhGASA* genes, 2 *PotriGASA* genes, and 1 *LOCOsGASA* gene. Subfamily CIV (Figure 1) contains 24 GASA genes, including 10 *GohirGASA* genes, 9 *AhGASA* genes, 2 *ATGASA* genes, 2 *PotriGASA* genes, and 1 *GlymaGASA* gene.

### 2.2. GASA Gene Expression Patterns in Different Peanut Tissues under Normal Conditions

According to the data collected from the Phytozome database, most of the AhGASA proteins showed subcellular localization in the extracellular space, nucleus, and mitochondria. In total, 40 GASA genes were identified in peanuts. According to the expression database profile results, we investigated 18 GASA genes (Appendix A), and selected genes were reconfirmed via qRT-PCR in certain tissues of peanut plants under conditions. Samples were collected during the seedling stage (young leaves, stems, and roots). We found that *AhGASA39*, *AhGASA31*, *AhGASA22*, *AhGASA25*, *AhGASA9*, *AhGASA19*, *AhGASA2*, *AhGASA30*, and *AhGASA27* exhibited significantly higher expression in leaves than in stems or roots under normal conditions (Figure 2). Similarly, *AhGASA1*, *AhGASA4*, *AhGASA21*, *AhGASA40*, *AhGASA13*, *AhGASA8*, *AhGASA29*, *AhGASA20*, and *AhGASA17* displayed significantly higher expression in stems than in leaves or roots (Figure 2). Interestingly, under normal conditions, *AhGASA21* and *AhGASA40* expression was significantly higher in roots than in leaves, whereas the other genes exhibited lower expression in roots than in stems or leaves (Figure 2). Most of the *AhGASA* genes exhibited high expression in young leaves and stems and lower expression in roots. A few of the *AhGASA* genes exhibited upregulated patterns under normal conditions, whereas some displayed downregulated or mixed patterns of expression (Figure 2).

### 2.3. Transcriptional Response of GASA Genes under Abiotic Stress (Drought and Salt)

To determine the possible role of *AhGASA* in abiotic stress, the relative expression of *AhGASA* genes was analyzed in control plants or plants exposed to drought or salt stress via qRT-PCR. Specifically, 11 genes (*XG1GJV_AHGASA39*, *VJ6NUA_AhG_ASA31*, *FK6HB9_AhGASA1*, *RUJ0QA_AhGASA22*, *VES4LD_AhGASA25*, *PW1K78_AhGASA19*, *TJLQ2T_AhGASA13*, *JGC45.1_AhGASA8*, *SY561U.1_AhGASA30*, *XBXR9Z.1_AhGASA29*, and *CH1EP2.1_AhGASA20*) exhibited significantly reduced transcript levels, 5 genes (*AVE3D2_AhGASA4*, *6SHE48_AhGASA9*, *FJ610L_AhGASA21*, *F1TUFL_AhGASA40*, and *C0SGC9.1_AhGASA27*) exhibited significantly higher transcript levels, and 1 gene (*YB2114_AhGASA2*) displayed no significant change in transcript levels under drought stress compared to the control findings (Figure 3).

Similarly, 11 genes (XG1GJV_AHGASA39, FK6HB9_AhGASA1, AVE3D2_AhGASA4, FJ610L_AhGASA21, F1TUFL_AhGASA40, PW1K78_AhGASA19, YB2114_AhGASA2, TJLQ2T_AhGASA13, SY561U.1_AhGASA30, C0SGC9.1_AhGASA27, and XBXR9Z.1_AhGASA2) exhibited significantly higher transcript levels under salt stress, whereas 5 genes (VJ6NUA_AhGASA31, RUJ0QA_AhGASA22, VES4LD_AhGASA25, AVE3D2_AhGASA4, and JGC45.1_AhGASA8) displayed significantly lower transcript levels (Figure 3). Among the 11 upregulated genes, 6 (XG1GJV_AHGASA39, FK6HB9_AhGASA1, PW1K78_AhGASA19, TJLQ2T_AhGASA13, SY561U.1_AhGASA30, and XBXR9Z.1_AhGASA29) exhibited different expression patterns between drought and salt stress. These results indicate that some *AhGASA* genes might play important roles in the regulation of salt stress, whereas others play important roles in the response to drought stress. Further functional characterization of upregulated genes under both drought and salt stress will help to confirm the regulatory roles of the aforementioned genes.

### 2.4. Effects of Drought and Salt Stress on Plant Traits

The effects of drought and salt stress on plant attributes, such as plant weight, shoot length, root length, and SPAD value (chlorophyll content), were determined after 2 weeks of stress, as presented in Appendix A. Under both abiotic stresses, plants exhibited significant reductions in plant weight, shoot length, root length, and total chlorophyll content compared to the findings in control plants (Appendix A). Interestingly, the root length and total chlorophyll content of salt-stressed plants were slightly higher than those of drought-stressed plants (Appendix A). Generally, plants subjected to abiotic stress modify their physiological mechanisms to combat this stress, resulting in alterations in attributes such as plant height, biomass, weight, shoot, and root length. Overall, these results indicated that drought and salt stress adversely affected the measured plant traits.

## 3. Discussion

Plant GASA proteins are widely reported in diverse plant species and well recognized for their vital roles in plant hormone signaling, plant organ growth and development, abiotic stress responses, and biotic stress defenses [22,23]. However, in many plant species, the functional roles of GASA genes are unknown. Studies have been conducted in plants subjected to different abiotic stresses to elucidate the regulatory involvement of GASA genes under abiotic stress [24,25,26]. A possible role in pod development was recently discovered for GASA genes in peanuts through expression analysis [8]. However, prior research did not evaluate GASA genes under abiotic stress. Thus, in this study, we analyzed the effects of drought and salt stress on plant traits and the expression of GASA genes in an effort to clarify their regulatory roles.

Phylogenetic evaluation of *AhGASA* in comparison to GASA genes from *Arabidopsis*, soybean, rice poplar, and cotton species revealed that most *AhGASA* genes are more closely related to soybean GASA genes, in line with the fact that peanuts and soybeans are both legumes. *AhGASA* genes are differentially distributed in four subfamilies. The phylogenetic results are consistent with previously reported studies of GASA genes in *Arabidopsis*, rice, poplar, tomato, soybean, and cotton [1,6,8,11,27,28].

The expression analysis revealed differences in the expression of GASA genes under normal conditions. The observation of specific gene expression in leaves, stems, and roots suggests that the genes have specific functions in the development of these organs or tissues. The expression of GASA genes in different tissues of cultivated peanuts provides a crucial hint for the discovery of their functional roles. Similarly, the tissue-specific expression of GASA genes was previously reported in cotton (*G. hirsutum*) [27]. However, the expression of some of the aforementioned genes exhibiting differential expression in peanut plant tissues needs to be validated in other peanut species.

The role of GASA genes in abiotic stress responses is well recognized. Recently, Bouteraa et al. (2023) [29] reported that the GASA gene in durum wheat (*TdGASA1*) was highly upregulated in response to salt and osmotic stress. Previously, a study in *Arabidopsis* revealed that the *AtGASA14* gene was involved in leaf expansion and salt stress tolerance through reactive oxygen species regulation [26]. An earlier overexpression study of *GASA4* in *Arabidopsis* suggested improved tolerance to heat stress in transgenic plants [24]. However, overexpression of *GASA5* in *Arabidopsis* uncovered a negative regulatory role in heat stress responses [30]. Another study in rice revealed the upregulation of GASA genes (*OsGASA1/8/10*) under salt stress [5]. These studies indicate that some GASA genes play vital positive or negative regulatory roles in abiotic stress responses. In line with previous studies, our investigation of GASA genes revealed differential expression patterns for selected genes in response to drought or salt stress. Furthermore, overexpression or complementation studies in a homologous or heterologous system for genes that are specifically upregulated or downregulated during drought or salt stress could provide precise insights into the functional roles of the genes.

The functional characterization of GASA genes has not been thoroughly completed in peanuts; thus, efforts to understand GASA genes via functional characterization studies in response to abiotic stress are needed. This will facilitate the development of peanut cultivars with enhanced tolerance to abiotic stresses such as drought and salt stress.

Typically, when plants are exposed to drought or salt stress, they exhibit morphological, physiological, molecular, and biochemical changes [31,32,33]. Previous studies reported that during the beginning of abiotic stress, the plant shoot length decreases immediately, whereas root growth is initially maintained before gradually declining, highlighting the adverse effects of stress on the entire plant. Studies have recorded an increased root/shoot ratio under abiotic stress in the early stage [34,35]. In our study, we observed that plants exposed to drought or salt stress exhibited significant reductions in plant weight, shoot length, root length, and chlorophyll content.

The impact of salt stress was more devastating on plant weight and shoot length than that of drought stress, whereas root length and chlorophyll content were more strongly affected by drought stress. Drought and salt stress result in reduced CO_2_ levels and suppressed photosynthesis, leading to a decrease in the RuBisCO content for carbon fixation and alterations in the enzymes involved in the biosynthesis of pigments, such as chlorophylls and carotenoids [36,37]. Our results are consistent with previously reported findings and revealed that drought and salt stress adversely affect plant morphological traits and chlorophyll content.

## 4. Materials and Methods

### 4.1. Experimental Condition

For this study, we chose a high-yielding variety ‘Palkwang’ seed materials specifically obtained from the Upland Crop Breeding Division, National Institute of Crop Science, Miryang (Gyeongnam, South Korea), and the experiment was conduct at this site. The seeds were sterilized using 70% ethanol, followed by multiple washes with diluted water. Afterward, the seeds were soaked in water and stored in a dark environment at 28 °C for 24 h to ensure consistent germination. Once the seeds germinated uniformly after 24 h, they were planted in a mixture of vermiculite, perlite, and peat moss (in a ratio of 2:2:1) in pots measuring 90 × 80 mm. Subsequently, the seedlings were transferred to a controlled greenhouse environment. The seedlings were grown under long-day conditions (16 h of light and 8 h of darkness) in a temperature range of 25 °C–28 °C.

Following the initial planting, regular watering was performed, and all plants were treated equally for 2 weeks. After this period, the two-week-old peanut plants in the greenhouse were divided into two sets. One set was subjected to water deprivation for 7 days, whereas the other set received ample water. The evaluation of plant growth, development, and chlorophyll content (soil–plant analysis development (SPAD) meter) was conducted on both well-watered and drought-stressed plants during the pot experiment.

Additionally, 2-week-old seedlings were exposed to salt stress using NaCl (150 mM) for 7 days. The concentration of 150 mM was based on a preliminary study and previous research conducted on legume species. Following salt stress, samples were collected and stored at −80 °C until further analysis as described for the drought stress experiment.

### 4.2. Identification of GASA Family Candidate Genes in A. hypogaea

To identify *AhGASA* genes, 15 GASA sequences from *Arabidopsis* (TAIR; http://www.arabidopsis.org accessed on 5 May 2022) were used for BLAST searches against the peanut genome in the Phytozome database (https://phytozome-next.jgi.doe.gov/report/transcript/Ahypogaea_v1_0 accessed on 5 May 2022). Consequently, putative GASA protein sequences were identified via a HMMER search against *A. hypogea*. Further, genomic DNA, CDS, transcripts, protein sequences, and 1500 bp sequences upstream of the promoter were obtained from the Phytozome database (https://phytozome-next.jgi.doe.gov/report/transcript/Ahypogaea_v1_0 accessed on 5 May 2022).

### 4.3. Extraction of RNA, cDNA Synthesis, and Quantitative Real-Time PCR (qRT-PCR) Analysis of GASA Genes

Expression analysis of selected GASA genes at different plant developmental growth stages (leaves, stems, and roots) and expression analysis of drought-stressed plants were conducted as previously described [38]. To quantify the expression of selected *AhGASA* genes, the actin gene was used as a reference housekeeping gene [39]. Peanut plant tissue was collected, frozen immediately in liquid nitrogen, and stored at −80 °C. Total RNA was extracted using an RNeasy PowerPlant Kit (Cat No. 13500-50), and quantification was performed using a NanoDrop spectrophotometer (Thermo Fisher Scientific, Waltham, MA, USA). A PrimeScript 1st strand cDNA synthesis kit (Takara, Shiga, Japan) was used according to the manufacturer’s protocol for reverse transcription. GASA gene-specific primers were designed using Primer 6.0 and examined using NCBI Primer-BLAST (https://www.ncbi.nlm.nih.gov/tools/primerblast/ accessed on 5 May 2022). qRT-PCR of selected GASA genes was performed at various developmental stages (leaves, stems, and roots) in stressed or unstressed samples as described previously [40]. The expression value was calculated via the relative expression method.

### 4.4. Phylogenetic Analysis

To investigate the evolutionary relationships of the GASA genes, a list of the GASA genes, including upregulated and downregulated genes, found in the transcriptome database was used as a query in Phytozome against *Arabidopsis* (*Arabidopsis Thaliana*), rice (*Oryza sativa*), soybean (*Glycine max*), poplar (*P. trichocarpa*), and cotton (*Gossypium hirsutum*). These orthologous protein sequences were obtained from the Phytozome database, and these sequences were aligned using ClustalW. Phylogenetic trees were constructed via the neighbor-joining method using MEGA software (http://www.megasoftware.net/ accessed on 5 May 2022), and a bootstrap test was performed with 1000 replicates. Based on the analysis, protein sequences were hierarchically clustered, and each cluster was manually separated into subclades.

## 5. Conclusions

In the study, we conducted a comprehensive genome-wide investigation of the GASA gene family in cultivated peanuts in an effort to uncover their evolutionary relationships and their expression patterns under normal conditions and under abiotic stress. This study reconfirmed the identity of 40 GASA genes in cultivated peanuts. Furthermore, through a biological experiment and the gene expression analysis of selected genes under abiotic stress and control conditions, we uncovered the potential regulation of GASA genes in response to drought and salt stress. This study provides vital information for further functional characterization of GASA genes to identify their exact regulatory roles in the context of abiotic stress responses. As peanut is an economically important crop for farmers that is cultivated widely as an oilseed crop, this study could improve our understanding of GASA genes for crop improvement and could lead to the development of resilient peanut varieties in view of challenging environmental conditions.

## Figures and Tables

**Figure 1 ijms-24-17117-f001:**
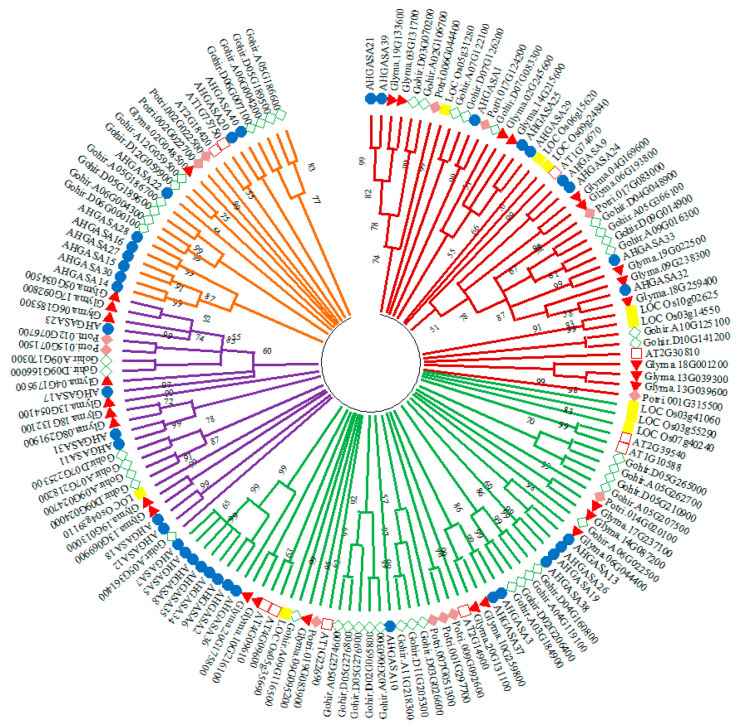
Phylogenetic tree of 148 GASA genes from different plant species (peanut, *Arabidopsis*, poplar, soybean, tomato, and cotton). The phylogenetic tree was created via the neighbor-joining method using MEGA X 64-bit version software (http://www.megasoftware.net/, accessed on 15 May 2022). Subfamilies are colored as follows: CI, red; CII, green; CIII, purple; and CIV, green. Different species are indicated using prefixes (AT, *A. thaliana*; Glyma, *G. max*; Potri, *P. trichocarpa*; Gohir, *G. hirsutum*; *LOC Os.*, *Oryza sativa*; and AH, *A. hypogaea*), followed by the GASA protein number.

**Figure 2 ijms-24-17117-f002:**
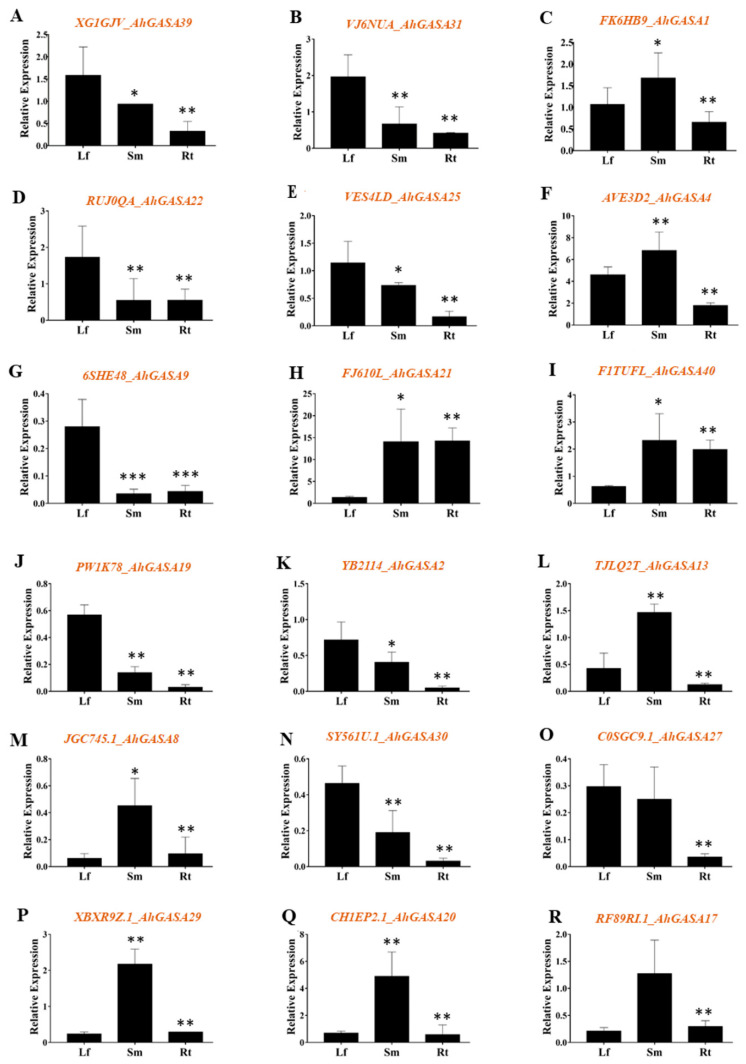
The relative expression of *AhGASA* genes in different tissues of peanut plants. The gene expression levels of (**A**) *AhGASA39*, (**B**) *AhGASA31*, (**C**) *AhGASA1*, (**D**) *AhGASA22*, (**E**) *AhGASA 25*, (**F**) *AhGASA4*, (**G**) *AhGASA9*, (**H**) *AhGASA21*, (**I**) *AhGASA40*, (**J**) *AhGASA19*, (**K**) *AhGASA2*, (**L**) *AhGASA13*, (**M**) *AhGASA8*, (**N**) *AhGASA30*, (**O**) *AhGASA27*, (**P**) *AhGASA29*, (**Q**) *AhGASA20* and (**R**) *AhGASA17.* Data are presented as the mean ± standard error of three biological replicates. Data were analyzed for significant differences using Student’s *t*-test. Asterisks (*) indicate significant differences at a 5% level (*** *p* < 0.001, ** *p* < 0.01, * *p* < 0.05).

**Figure 3 ijms-24-17117-f003:**
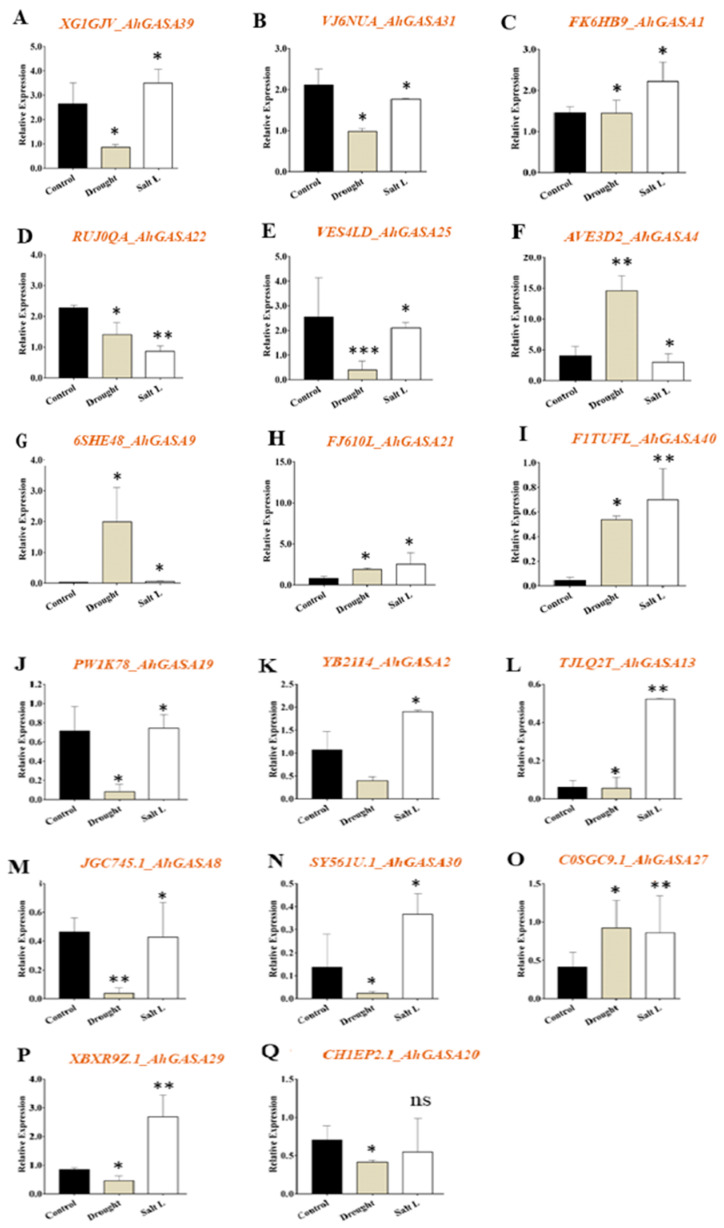
The relative levels of peanut *AhGASA* genes grown under drought, salinity, and control conditions. The gene expression levels of (**A**) *AhGASA39*, (**B**) *AhGASA31*, (**C**) *AhGASA1*, (**D**) *AhGASA22*, (**E**) *AhGASA25*, (**F**) *AhGASA4*, (**G**) *AhGASA9*, (**H**) *AhGASA21*, (**I**) *AhGASA40*, (**J**) *AhGASA19*, (**K**) *AhGASA2*, (**L**) *AhGASA13*, (**M**) *AhGASA8*, (**N**) *AhGASA30*, (**O**) *AhGASA27*, (**P**) *AhGASA29*, and (**Q**) *AhGASA20*. Data are presented as the mean ± standard error of three biological replicates. Means were analyzed for significant differences using Student’s *t*-test. Asterisks (*) indicate significant differences at a 5% level (*** *p* < 0.001, ** *p* < 0.01, * *p* < 0.05).

## Data Availability

Data are contained within the article and Appendix A.

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
