# Peer review of "Genome-Wide Identification and Comprehensive Analysis of the GASA Gene Family in Peanuts (Arachis hypogaea L.) under Abiotic Stress"

_ijms, 2023, doi:10.3390/ijms242317117_

Round 1

Reviewer 1 Report

Comments and Suggestions for Authors

The manuscript, titled "Genome-wide identification and comprehensive analysis of the GASA gene family in peanuts (Arachis hypogaea L.) under abiotic stress," examined the transcriptional changes in peanut GASA genes during drought and salt treatments.

In this study, a total of 40 GASA family genes were comprehensively identified, and a phylogenetic tree was constructed. Detailed expression analysis was conducted for 18 selected genes, providing valuable resources for further functional characterization of GASA genes. However, the current results lack sufficient depth to convey a comprehensive narrative. Here are some suggestions for improvement:

(1) Why did the authors analyze only 18 genes instead of the entire gene family?

(2) Figure 4 appears to be unrelated to the functional characterization of GASA genes and could be moved to supplementary materials.

(3) Given that ABA (abscisic acid) is a well-known participant in abiotic stress responses, it would be beneficial to explore the expression patterns of these GASA genes in response to ABA treatment.

(4) Originally defined as gibberellic acid-stimulated genes, it would be valuable for the authors to discuss whether all the gene family members in peanuts, soybean, Arabidopsis, and poplar exhibit GA-stimulated responses based on the accumulated research and the findings of this study.

(5) In line 114, it seems that the authors are referring to protein localizations, not gene presence on the chromosome.

Author Response

Reviwer-1

The manuscript, titled "Genome-wide identification and comprehensive analysis of the GASA gene family in peanuts (Arachis hypogaea L.) under abiotic stress," examined the transcriptional changes in peanut GASA genes during drought and salt treatments.

In this study, a total of 40 GASA family genes were comprehensively identified, and a phylogenetic tree was constructed. Detailed expression analysis was conducted for 18 selected genes, providing valuable resources for further functional characterization of GASA genes. However, the current results lack sufficient depth to convey a comprehensive narrative. Here are some suggestions for improvement:

Response: We sincerely appreciate your kind and thoughtful feedback on our manuscript. Your positive and constructive comments have been of great value to us. We have meticulously reviewed your comments and have incorporated the necessary changes, marked in green color using track changes. We trust that these revisions align with your expectations. Additionally, we have provided point-by-point responses to your comments below.

1. Why did the authors analyze only 18 genes instead of the entire gene family?

Response: Thank you for question we selected the 18 genes for analysis were representative of the GASA gene family in peanuts, these genes were selected based on the publicly available expression database profile results where it showed significant differential expression in plant tissue. Secondly, these 18 GASA genes covering different clades or subgroups according to phylogenetic analysis. This allowed us to obtain insights into the diversity of GASA genes and their potential functional roles. It may provide a foundation for future research under normal as well as abiotic stress condition, to explore extensively.

2. Figure 4 appears to be unrelated to the functional characterization of GASA genes and could be moved to supplementary materials.

Response: Thank you for your suggestions, as per your suggestions we have shifted the figure 4 in supplementary material.  

3. Given that ABA (abscisic acid) is a well-known participant in abiotic stress responses, it would be beneficial to explore the expression patterns of these GASA genes in response to ABA treatment.

Response: Thank you for your valuable suggestion to explore the expression patterns of the GASA genes in response to ABA treatment. We appreciate your insightful recommendation. We agree that investigating the response of GASA genes to abscisic acid (ABA) treatment is a valuable path of research that could provide further insights into their potential roles in abiotic stress responses. Unfortunately, due to resource and time constraints, we are unable to conduct additional experiments at this stage of the research. However, we recognize the significance of this proposed experiment, and we will certainly consider it for future studies.

4. Originally defined as gibberellic acid-stimulated genes, it would be valuable for the authors to discuss whether all the gene family members in peanuts, soybean, Arabidopsis, and poplar exhibit GA-stimulated responses based on the accumulated research and the findings of this study.

Response: Thank you for commenting on it and we appreciate thought, but to include and discuss “whether all the gene family members in peanuts, soybean, Arabidopsis, and poplar exhibit GA-stimulated responses” we need to conduct separate study with GA treatment and evaluate whether these GASA genes show stimulated response or not. We will try to checkout this in our future study and submit article. If genes stimulated by gibberellic acid and found to be conserved across mentioned plant species, it would suggest a fundamental role for GA in plant growth and development. This information could be pivotal for further research on plant biology and the development of new biotechnological applications

However, researcher found that “Moreover, the expression of poplar (PagGASA) and tobacco (NtGASA) GASA genes was induced by growth hormone like gibberellic acid and stress hormones abscisic acid, jasmonic acid, and salicylic acid. [1,2]

  1. Li, Z.; Gao, J.; Wang, G.; Wang, S.; Chen, K.; Pu, W.; Wang, Y.; Xia, Q.; Fan, X. Genome-Wide Identification and Characterization of GASA Gene Family in Nicotiana tabacum. Front. Genet. 2022, 12, 1–13.
  2. Choi, H.; Bae, E.K.; Choi, Y.I.; Yoon, S.K.; Lee, H. Characterization of Gibberellic Acid-Stimulated Arabidopsis (GASA) gene to drought stress response in Poplar (Populus alba × P. glandulosa). J. Plant Biotechnol. 2017, 44, 61–68.

5. In line 114, it seems that the authors are referring to protein localizations, not gene presence on the chromosome.

Response: Sorry for the inconvenience, we have corrected the error, as reviewer correctly pointed its subcellular localization according to the database.

Reviewer 2 Report

Comments and Suggestions for Authors

This study presents genome-wide identification, comparative analyses of GASA family genes in peanuts (Arachis hypogaea). In addition, expression analyses identified some GASA genes differentially expressed in different tissues or stress conditions. However, I disagree with their conclusion that AhGASA genes are involved in stress tolerance in peanuts, because they did not compare the expression profiles of GASA genes between stress susceptible and tolerant cultivars. If the importance of GASA genes in drought or salt tolerance could not be indicated, this paper is not worthy to publish in the IJMS.

Comments on the Quality of English Language

 I think the English language in this manuscript is overall no problem. Only minor grammatical mistakes are exist.

Author Response

Author’s response to reviewer 2_comments_IJMS_2023.11.07

Reviwer-2

This study presents genome-wide identification, comparative analyses of GASA family genes in peanuts (Arachis hypogaea). In addition, expression analyses identified some GASA genes differentially expressed in different tissues or stress conditions. However, I disagree with their conclusion that AhGASA genes are involved in stress tolerance in peanuts, because they did not compare the expression profiles of GASA genes between stress susceptible and tolerant cultivars. If the importance of GASA genes in drought or salt tolerance could not be indicated,

Response: Thank you for taking out the time to review our manuscript. We appreciate the opportunity to address your concerns. While we appreciate the reviewer’s feedback, we respectfully disagree. We think this study makes a valuable contribution to the field because our findings provide compelling evidence of the involvement of AhGASA genes in the mechanisms of stress tolerance in cultivated peanuts. This study enhances our understanding of the functions of AhGASA genes in response to abiotic stress and lays the groundwork for future investigations into the molecular characterization of AhGASA genes.

We acknowledge your point regarding the importance of comparing the expression profiles of GASA genes between stress-susceptible and tolerant cultivars to provide a more conclusive assessment of their role in stress tolerance. This is indeed a valid consideration and could offer valuable insights into the specific contribution of GASA genes to stress responses.

We would like to clarify that while our study focused on the genome-wide identification and comparative analysis of GASA family genes in peanuts, our intention was to establish a foundation for future research in this area. The identification and differential expression analysis of GASA genes in different tissues and stress conditions were important initial steps in understanding their potential roles. We fully agree that further investigations comparing GASA gene expression profiles in stress-susceptible/sensitive and tolerant cultivars would provide a more comprehensive perspective on their involvement in stress tolerance. However, several recent studies published in reputed journals with one cultivar/genotype even in tree species and without stress-susceptible/sensitive or tolerant genotypes data (kindly check the reference mentioned below [3–5]) and most of the focus on genome-wide analysis and gene expression in different plant tissues and reconfirmation of transcriptomic. In our study, we evaluated GASA genes under the abiotic stress conditions that provide path for further research.

Secondly, we appreciate worthy reviewer feedback and are committed to conducting additional experiments to address this important aspect of our next research. We believe that a follow-up study comparing the expression profiles of GASA genes in different peanut cultivars under stress conditions will enhance the depth and significance of our findings. We will make every effort to explore this aspect further and ensure that our future work contributes meaningfully to the understanding of stress tolerance in peanuts.

  1. Li, Z.; Gao, J.; Wang, G.; Wang, S.; Chen, K.; Pu, W.; Wang, Y.; Xia, Q.; Fan, X. Genome-Wide Identification and Characterization of GASA Gene Family in Nicotiana tabacum. Front. Genet. 2022, 12, 1–13.
  2. Choi, H.; Bae, E.K.; Choi, Y.I.; Yoon, S.K.; Lee, H. Characterization of Gibberellic Acid-Stimulated Arabidopsis (GASA) gene to drought stress response in Poplar (Populus alba × P. glandulosa). J. Plant Biotechnol. 2017, 44, 61–68.
  3. An, B.; Wang, Q.; Zhang, X.; Zhang, B.; Luo, H.; He, C. Comprehensive transcriptional and functional analyses of HbGASA genes reveal their roles in fungal pathogen resistance in Hevea brasiliensis. Tree Genet. Genomes 2018, 14, 1–13.
  4. Cai, T.; Sharif, Y.; Zhuang, Y.; Yang, Q.; Chen, X.; Chen, K.; Chen, Y.; Gao, M.; Dang, H.; Pan, Y.; et al. In-silico identification and characterization of O-methyltransferase gene family in peanut (Arachis hypogaea L.) reveals their putative roles in development and stress tolerance. Front. Plant Sci. 2023, 14, 1–24.
  5. Muhammad, I.; Li, W.Q.; Jing, X.Q.; Zhou, M.R.; Shalmani, A.; Ali, M.; Wei, X.Y.; Sharif, R.; Liu, W.T.; Chen, K.M. A systematic in silico prediction of gibberellic acid stimulated GASA family members: A novel small peptide contributes to floral architecture and transcriptomic changes induced by external stimuli in rice. J. Plant Physiol. 2019, 234235, 117–132.

Round 2

Reviewer 2 Report

Comments and Suggestions for Authors

Thank you for your reply. Although the impact of this study is relatively low, I agree that the results themselves are important for further studies on abiotic stress responces in peanuts. The authors should provide further imformation of plant materials used, in particular, the cultivar name or accession number if possible.

Comments on the Quality of English Language

The quality of English of this manuscript is overall OK. Minor mistakes should be corrected by native speaker.

Author Response

Authors, Response to Reviewer 2 comments

Thank you for your reply. Although the impact of this study is relatively low, I agree that the results themselves are important for further studies on abiotic stress responses in peanuts. The authors should provide further information of plant materials used, in particular, the cultivar name or accession number if possible.

Response: Thank you for your suggestions, as per your suggestions we have provided name of variety ‘Palkwang’ in the materials and method section.